# Clinical characteristics and outcomes of 952 hospitalized COVID-19 patients in The Netherlands: A retrospective cohort study

Niels Pouw[1], Josephine van de Maat[1], Karin Veerman[2], Jaap ten Oever[1], Nico Janssen[1], Evertine Abbink[1], Monique Reijers[3], Quirijn de Mast[1], Wouter Hoefsloot[1], Reinout van Crevel[1], Kitty Slieker[4], Marjan van Apeldoorn[5], Marc Blaauw[1,4], Anton Dofferhoff[6], Jacobien Hoogerwerf[1]*

1 Department of Internal Medicine and Radboud Center for Infectious Diseases, Radboud University Medical Center, Nijmegen, The Netherlands, 2 Department of Internal Medicine, St. Maartenskliniek, Nijmegen, The Netherlands, 3 Department of Pulmonary Diseases, Radboud University Medical Center, Nijmegen, The Netherlands, 4 Department of Internal Medicine, Bernhoven Hospital, Uden, The Netherlands, 5 Department of Internal Medicine, Jeroen Bosch Hospital, 's-Hertogenbosch, The Netherlands, 6 Department of Internal Medicine, Canisius Wilhelmina Hospital, Nijmegen, The Netherlands

* Jacobien.Hoogerwerf@radboudumc.nl

## Abstract

### Objective

To describe clinical characteristics, disease course and outcomes in a large and well-documented cohort of hospitalized COVID-19 patients in the Netherlands.

### Methods

We conducted a multicentre retrospective cohort study in The Netherlands including 952 of 1183 consecutively hospitalized patients that were admitted to participating hospitals between March 2nd, 2020, and May 22nd, 2020. Clinical characteristics and laboratory parameters upon admission and during hospitalization were collected until July 1st.

### Results

The median age was 69 years (IQR 58–77 years) and 605 (63.6%) were male. Cardiovascular disease was present in 558 (58.6%) patients. The median time of onset of symptoms prior to hospitalization was 7 days (IQR 5–10). A non ICU admission policy was applicable in 312 (32.8%) patients and in 165 (56.3%) of the severely ill patients admitted to the ward. At admission and during hospitalization, severely ill patients had higher values of CRP, LDH, ferritin and D-dimer with higher neutrophil counts and lower lymphocyte counts. Overall in-hospital mortality was 25.1% and 183 (19.1%) patients were admitted to ICU, of whom 56 (30.6%) died. Patients aged ≥70 years had high mortality, both at the ward (52.4%) and ICU (47.4%). The median length of ICU stay was 8 days longer in patients aged ≥70 years compared to patients aged ≤60 years.

**Data Availability Statement:** All relevant data are within the paper and its Supporting information Files.

**Funding:** The authors received no specific funding for this work.

**Competing interests:** The authors have declared that no competing interests exist.

## Conclusion

Hospitalized COVID-19 patients aged ≥70 years had high mortality and longer ICU stay compared to patients aged ≤60 years. These findings in combination with the patient burden of an ICU admission and possible long term complications after discharge should encourage us to further investigate the benefit of ICU admission in elderly and fragile COVID-19-patients.

## Introduction

In March 2020, the World Health Organization declared coronavirus disease 2019 (COVID-19) a pandemic [1]. On February 6th, 2021 over 100 million confirmed cases have been reported worldwide, including over 2 million deaths [2]. The first laboratory-confirmed case of COVID-19 in the Netherlands was reported on February 27, 2020 [3]. On February 6th, 2021, close to a million confirmed cases have been reported in the Netherlands, including 14.234 deaths [2].

Disease severity ranges from mild flu-like symptoms to severe pneumonia and acute respiratory distress syndrome [4–6], requiring mechanical ventilation in a significant number of patients [7]. Several studies have already described details at admission such as clinical characteristics, demographics and laboratory parameters. Furthermore, several studies reported complications and clinical outcomes. However, results were often biased by the fact that many patients were still in hospital at the time of publication [4, 5, 7–13]. Therefore, more solid data on complications and outcomes are required. Moreover, insights into laboratory parameters is necessary to improve diagnostic and therapeutic management, especially in patients with high risk of poor outcome.

Around the world, there is wide regional variation in ICU admission policies [14]. In The Netherlands, ICU admission policies are commonly discussed between a physician and a patient and registered at the time of hospital admission. During the first COVID-19 wave in the Netherlands, in times of movement restrictions and a (partial) lockdown, physicians were encouraged to discuss the ICU admission policy with their patients [15]. This decision depends on the general condition of the patient (and thereby the expected benefit of an ICU admission) and most importantly, the patient's preference. As a result, we expected a high number of severely ill patients who were admitted to the ward with a non ICU admission policy in our population.

The objective of this large cohort study is to describe and differentiate the clinical features, laboratory parameters during the course of illness, complications and clinical outcomes between severely and moderately ill patients hospitalized with COVID-19 in The Netherlands.

## Methods

### Study design and population

In this multicenter cohort study, all consecutive patients admitted with SARS-CoV-2 infection to five hospitals in The Netherlands were screened for eligibility between March 2th 2020 and May 22th 2020. All hospitals were located in the province of Gelderland and North-Brabant. Patients were included if they had at least 1 positive SARS-CoV-2 polymerase-chain-reaction (PCR) test of a nasopharyngeal or throat swab or sputum specimen [16]. For this study,

patients were excluded if follow-up details (e.g. reason of discharge) were missing or if patients were still hospitalized at the time of database closure.

## Study procedures and data collection

Routine care data from included patients were extracted from the electronic medical records (EMR) and entered anonymously into a web-based electronic case report form (eCRF; Castor Electronic Data Capture). The CRF for this study was based on the ISARIC-WHO COVID-19 CRF. Medical records were accessed between March 17th and the 1st of July. The following data were collected: baseline characteristics, demographics, admission details, co-morbidities, symptoms at presentation, vital signs, laboratory parameters, radiological findings (i.e. chest CT and X-ray), supportive care and medication during admissions, complications and clinical outcomes (e.g. death, discharge, transfer). For admission data, first available values recorded in the Emergency Department were used. Laboratory tests were performed in the laboratories of admitting hospitals. COVID-19 Reporting and Data System (CO-RADS) score was used to identify the likelihood of COVID-19 infection based on a CT-scan [17]. Furthermore, vital signs were collected on admission and laboratory parameters were collected 3–4 times per week during hospitalization. If data were missing, this is indicated in the tables. For laboratory values and vital signs on admission, the first documented value was taken. Clinical outcome, supportive care, complications and treatment during hospital stay were evaluated at the time of death or discharge. Follow-up of clinical outcomes was halted on July 1st.

## Main outcomes

Characterization of patients, including demographic and clinical characteristics, presenting symptoms, comorbidities, vital signs, laboratory parameters upon admission and during hospitalization, COVID-19 radiologic, disease severity and clinical outcome data (e.g. mortality).

## Definitions

Moderately and severely ill patients were defined according to the WHO-definition of COVID-19 disease severity [18]. Patients were considered severely ill if they had clinical signs of pneumonia and they met 1 of the following requirements: a respiratory rate of >30, sPO2 <90 or severe respiratory distress requiring ICU admission, a Venturi mask or non-rebreather mask. Moderately ill patients were all other patients that were hospitalized in this cohort, but did not meet the requirements of severely ill patients stated above. Age was stratified in three different subgroups: <60 years, 61–69 years and >70 years. Immunocompromised status was defined as hematological malignancies, auto-immune diseases and/or HIV/AIDS or treatment with stem cell or organ transplantations and/or use of immunosuppressive medication (corticosteroids, monoclonal antibodies interfering with the immune system, small molecular immunosuppressant or antineoplastic agents) at admission. Solid organ malignancies without current use of immunosuppressive medication were assumed medical history rather than active malignancies in the majority of patients, and thus were not included in the definition of 'immunocompromised'. Disease onset was defined as the first day of symptoms. Obesity was defined as a body mass index (BMI) greater than 30 kg/m$^2$. Liver function abnormalities were defined as aspartate aminotransferase (ASAT) or alanine aminotransferase (ALAT) levels five times above the upper limit of normal. Acute kidney injury (AKI) was defined as an eGFR < 60 ml/min. or >25% decrease in eGFR [19]. The CT severity score was used to evaluate the severity of pulmonary involvement in COVID-19 patients, based on a scale from 0–40 [20]. Hyperglycemia was defined as a blood glucose level requiring insulin therapy. Hypoglycemia was defined as blood glucose levels lower than 2.2 mmol/L.

### Ethical considerations

This study was not subject to the Medical Research Involving Human Subjects Act (WMO) in the Netherlands, and has been reviewed by the institutional review board of the Radboud university medical center (number 2020–2923 and 2020–6344). According to the institutional review board, only oral consent was required. Oral consent was obtained from all patients or their family and documented in the electronic medical records.

### Statistical analysis

We used descriptive statistics to report the clinical characteristics, diagnostics and disease course of our population. Continuous variables were expressed as medians including inter-quartile ranges (IQR) and as means including standard deviations. Chi-squared tests were used for dichotomous and categorical variables and t-tests for continuous variables to compare the characteristics of severely ill versus moderately ill patients. Two-sided 5% significance levels were used to identify statistically significant results. All confidence intervals reported are 95% confidence intervals. We used SPSS version 25.0 (IBM SPSS Statistics) for the descriptive analyses. R Statistics (version 3.6.3) was used for constructing timeseries graphs of laboratory values from disease onset and expressing survival of different age subgroups in a Kaplan-Meier curve.

## Results

### Demographics and comorbidities

In total, 1183 hospital admitted suspected COVID-19 patients were identified, of whom 99 patients did not give informed consent (S1 Fig in S1 Data). Of the remaining 1084 patients, 49 patients did not have a PCR-confirmed SARS-CoV-2. After excluding 83 patients due to no follow up details (e.g. reason of discharge) from our total study population (n = 1035), a total of 952 patients with laboratory-confirmed SARS-CoV-2 infection in the Netherlands were included in this study.

Among the 952 patients, 605 (63.6%) were male and median age was 69 years (IQR 58–77), see Table 1. The most common comorbidity was cardiovascular disease (558 [58.6%]), including hypertension (374 [39.3%]). Pulmonary disease (238 [25.0%]), diabetes mellitus (215 [22.6%]) and an immunocompromised state (204 [21.4%]) were other frequently observed comorbidities.

In total, 476 (50.0%) patients were moderately ill and 476 (50.0%) were severely ill. Severely ill patients were predominantly males (68.3% versus 58.8%; p = 0.002) and had a history of myocardial infarction more frequently than moderately ill patients (11.6% versus 7.4% p = 0.027). There was no significant age difference between severely and moderately ill patients (70 (IQR 56–78) versus 69 (IQR 61–77); p = 0.113). Obesity and immunocompromised status were equally prevalent in both groups.

### Reported symptoms and vital signs

The median duration of symptoms prior to hospitalization was 7 days (IQR 5–10). The three most commonly reported symptoms were fever (720[76.1%]), cough (716[75.7%]) and dyspnea (660[69.8%]) (Table 1). Other reported symptoms were: fatigue (330[34.9%]), diarrhea (312[33.0%]), nausea or vomiting (239[25.3%]), headache (236, [21.5%]) and muscle aches (179[16.2%]).

Moderately ill patients suffered more often from nausea or vomiting and headache than severely ill patients (29.0% vs 21.5%; p<0.036 and 24.6% vs 18.5%; p<0.023, respectively).

**Table 1. Baseline demographics, clinical characteristics and vital signs in moderately and severely ill COVID-19 patients at admission.**

| | Total Patients (n = 952) | Moderately ill patients (n = 476) | Severely ill patients (n = 476) | P Value[a] | Missing values n (%) |
|---|---|---|---|---|---|
| **Demographics–***no. (%)* | | | | | |
| Gender, male | 605 (63.6) | 280 (58.8) | 325 (68.3) | **0.002** | 0 (0) |
| Median age:–*(IQR)* | 69 (58–77) | 69 (56–78) | 70 (61–77) | 0.113 | 0 (0) |
| *<60 years* | 274 (28.8) | 157 (33.0) | 117 (24.6) | **0.004** | 0 (0) |
| *61–69 years* | 205 (21.5) | 92 (19.3) | 113 (23.7) | 0.098 | 0 (0) |
| *>70 years* | 473 (49.7) | 227 (47.7) | 246 (51.7) | 0.218 | 0 (0) |
| Obesity[b] –*no./total no. of patients (%)* | 213/685 (31.1) | 106/320 (33.1) | 107/365 (29.3) | 0.282 | 267 (28.0) |
| **Comorbidities—***no. (%)* | | | | | |
| Diabetes mellitus | 215 (22.6) | 108 (22.7) | 107 (22.5) | 0.938 | 0 (0) |
| Pulmonary disease | 238 (25.0) | 121 (25.4) | 117 (24.6) | 0.765 | 0 (0) |
| Cardiovascular disease | 558 (58.6) | 268 (56.3) | 290 (60.9) | 0.148 | 0 (0) |
| *Hypertension* | 374 (39.3) | 179 (37.6) | 195 (41.0) | 0.288 | 0 (0) |
| *Myocardial infarction* | 90 (9.5) | 35 (7.4) | 55 (11.6) | **0.027** | 0 (0) |
| *Atrial fibrillation* | 136 (14.3) | 70 (14.7) | 66 (13.9) | 0.711 | 0 (0) |
| *Heart failure* | 51 (5.4) | 22 (4.6) | 29 (6.1) | 0.314 | 0 (0) |
| Cerebrovascular disease | 117 (12.3) | 56 (11.8) | 61 (12.8) | 0.622 | 0 (0) |
| Liver disease | 26 (2.7) | 17 (3.6) | 9 (1.9) | 0.112 | 0 (0) |
| Solid organ malignancy | 140 (14.7) | 60 (12.6) | 80 (16.8) | 0.067 | 0 (0) |
| Hematological malignancy | 41 (4.3) | 21 (4.4) | 20 (4.2) | 0.873 | 0 (0) |
| Solid organ transplantation | 14 (1.5) | 9 (1.9) | 5 (1.1) | 0.281 | 0 (0) |
| Stem cell transplantation | 7 (0.7) | 5 (1.1) | 2 (0.4) | 0.255 | 0 (0) |
| Chronic kidney disease requiring RRT | 13 (1.4) | 8 (1.7) | 5 (1.1) | 0.402 | 0 (0) |
| Chronic kidney disease not requiring RRT | 96 (10.1) | 52 (10.9) | 44 (9.2) | 0.389 | 0 (0) |
| Auto-immune disease, including IBD | 117 (12.3) | 65 (13.7) | 52 (10.9) | 0.199 | 0 (0) |
| Immunocompromised | 204 (21.4) | 109 (22.9) | 95 (20.0) | 0.269 | 0 (0) |
| **Reported symptoms–***no. (%)* | | | | | |
| Median onset of symptoms prior to hospitalization–*(IQR)* | 7 (5–10) | 7 (5–10) | 7 (5–10) | 0.928 | 81 (8.5) |
| Fever (>38,0 C) | 720 (76.1) | 356 (74.8) | 364 (77.4) | 0.338 | 6 (0.6) |
| Chills | 115 (12.2) | 66 (13.9) | 49 (10.4) | 0.105 | 6 (0.6) |
| Cough (with or without sputum) | 716 (75.7) | 356 (74.8) | 360 (75.6) | 0.764 | 6 (0.6) |
| Fatigue | 330 (34.9) | 175 (36.8) | 155 (33.0) | 0.222 | 6 (0.6) |
| Hemoptysis | 15 (1.6) | 7 (1.5) | 8 (1.7) | 0.776 | 6 (0.6) |
| Dyspnea | 660 (69.8) | 306 (64.3) | 354 (75.3) | **<0.001** | 6 (0.6) |
| Chest pain | 114 (12.1) | 62 (13.0) | 52 (11.1) | 0.354 | 6 (0.6) |
| Rhinorrhea | 51 (5.4) | 30 (6.3) | 21 (4.5) | 0.212 | 6 (0.6) |
| Sore throat | 37 (3.9) | 20 (4.2) | 17 (3.6) | 0.643 | 6 (0.6) |
| Loss of smell or taste | 61 (6.4) | 37 (7.8) | 24 (5.1) | 0.095 | 6 (0.6) |
| Nausea or vomiting | 239 (25.3) | 138 (29.0) | 101 (21.5) | **0.008** | 6 (0.6) |
| Abdominal pain | 99 (10.5) | 57 (12.0) | 42 (8.9) | 0.127 | 6 (0.6) |
| Diarrhea | 312 (33.0) | 158 (33.2) | 154 (32.8) | 0.889 | 6 (0.6) |
| Headache | 204 (21.6) | 117 (24.6) | 87 (18.5) | **0.023** | 6 (0.6) |
| Altered consciousness / confusion | 64 (6.8) | 36 (7.6) | 28 (6.0) | 0.326 | 6 (0.6) |
| Muscle aches | 143 (15.1) | 69 (14.5) | 74 (15.7) | 0.592 | 6 (0.6) |
| Joint pain | 8 (0.8) | 5 (1.1) | 3 (0.6) | 0.489 | 6 (0.6) |
| Skin rash | 5 (0.5) | 3 (0.6) | 2 (0.4) | 0.664 | 6 (0.6) |
| **Vital signs at admission–***mean (SD)* | | | | | |

*(Continued)*

**Table 1.** (Continued)

| | Total Patients (n = 952) | Moderately ill patients (n = 476) | Severely ill patients (n = 476) | P Value[a] | Missing values n (%) |
|---|---|---|---|---|---|
| Systolic blood pressure (mmHg) | 134 (20.7) | 134 (20.5) | 134 (20.9) | 0.969 | 6 (0.6) |
| Diastolic blood pressure (mmHg) | 76 (13.8) | 77 (12.8) | 76 (14.7) | 0.053 | 18 (1.9) |
| Heart rate (bpm) | 93 (19.6) | 91 (19.4) | 95 (19.7) | **0.009** | 20 (2.1) |
| Respiratory rate—median (IQR) | 22 (18–27) | 22 (18–24) | 25 (20–31) | **<0.001** | 39 (4.1) |
| Median peripheral oxygen saturation level in percentage–*(IQR)* | 94 (92–96) | 95 (93–97) | 93 (89–96) | **<0.001** | 15 (1.6) |
| **qSOFA score**–*no./total no. of patients (%)* | | | | **<0.001** | 33 (3.5) |
| *0* | 382/919 (41.6) | 227/475 (47.8) | 155/444 (34.9) | - | - |
| *1* | 494/919 (53.8) | 240/475 (50.5) | 254/444 (57.2) | - | - |
| *2* | 40/919 (4.2) | 8/475 (1.7) | 32/444 (7.2) | - | - |
| *3* | 3/919 (0.3) | 0/475 (0) | 3/444 (0.7) | - | - |

Abbreviations: COVID-19, coronavirus disease 2019; IQR, interquartile range; BMI, body mass index; RRT, renal replacement therapy; IBD, inflammatory bowel disease; bpm, beats per minute; qSOFA, quick sequential organ failure assessment.

[a] P values for comparisons between moderately and severely ill patients. P < 0.05 was considered statistically significant.

[b] Obesity is defined as a BMI of >30 kg/m$^2$.

Dyspnea was more frequently seen in severely ill patients than in those who were moderately ill (75.3% versus 64.3%).

Severely ill patients, had a higher heart rate (95 bpm versus 91 bpm; p = 0.009), higher respiratory rate (25 (IQR 20–31) versus 22 (IQR 18–24); p = <0.001) and a lower peripheral oxygen saturation level (93 (IQR 89–96) versus 95 (93–97); p = <0.001) at admission (Table 1).

## Laboratory parameters, radiological findings at admission and treatment

Neutrophil count, C-reactive protein (CRP), ferritin, D-dimer, ALT, AST, lactate dehydrogenase (LDH), procalcitonin and serum creatinine were significantly higher in severely ill patients compared to moderately ill patients, whereas the lymphocyte count was significantly lower (Table 2).

In 570 (59.9%) patients a chest X-ray was performed and in 489 (51.4%) patients a chest CT scan. The median CT severity score, determined in 336 patients, was significantly higher in severely ill versus moderately ill patients (13 [IQR 10–17] and 10 [IQR 7–13], respectively; p<0.001). Most patients (76%) had a CORADS score of 5 at admission. In this cohort, 664 (69.7%) patients were treated predominantly with chloroquine. Hydroxychloroquine, Remdsivir, lopinavir/ritanovir and Anakinra were used scarcely at the time. A quarter of all patients received no medication.

## Course of laboratory parameters during hospitalization

Six different laboratory parameters were plotted relative to onset of symptoms prior to hospitalization in severely ill and moderately ill patients (Fig 1). CRP, LDH, neutrophils, ferritin and D-dimer were consistently higher in severely ill patients compared to moderately ill patients, whereas the lymphocyte count was lower in severely ill patients. After initial similar concentrations, D-dimer concentrations diverged from day 11 onwards, with highest values in the severely ill group.

**Table 2. Laboratory parameters and diagnostic findings at admission and treatment during hospital stay in moderately and severely ill COVID-19 patients.**

| Characteristic | Total Patients (n = 952) | Moderately ill patients (n = 476) | Severely ill patients (n = 476) | P Value[a] | Missing values, n (%) |
|---|---|---|---|---|---|
| **Laboratory findings at admission**–*Median (IQR)* | | | | | |
| Hemoglobin (mmol/l) | 8.5 (7.8–9.2) | 8.5 (7.8–9.1) | 8.6 (7.7–9.2) | 0.406 | 23 (2.4) |
| White blood cell count (x10^9) | 6.7 (5.0–9.1) | 6.5 (4.7–8.4) | 6.9 (5.1–9.9) | **0.003** | 23 (2.4) |
| Neutrophil count (x10^9) | 5.2 (3.6–7.3) | 4.9 (3.3–6.7) | 5.6 (3.8–8.3) | <**0.001** | 77 (8.1) |
| Lymphocyte count (x10^9) | 0.9 (0.6–1.2) | 0.90 (0.70–1.30) | 0.80 (0.60–1.10) | <**0.001** | 74 (7.8) |
| Neutrophil-to-lymphocyte ratio | 6.0 (3.7–9.9) | 5.1 (3.3–8.5) | 6.8 (4.2–11.5) | <**0.001** | 78 (8.2) |
| Thrombocyte count (x10^9) | 205 (159–270) | 206 (164–268) | 205 (153–272) | 0.418 | 31 (3.3) |
| C-reactive protein (CRP) (mg/l) | 91 (47–146) | 67 (37–119) | 109 (61–171) | <**0.001** | 21 (2.2) |
| Ferritin (μgram/l) | 785 (406–1443) | 647 (356–1199) | 951 (503–1686) | <**0.001** | 192 (20.2) |
| D-dimer (ng/ml) | 860 (400–1740) | 750 (350–1445) | 1010 (500–2190) | <**0.001** | 163 (17.1) |
| Total bilirubin (μmole/l) | 9.0 (7.0–13.0) | 9 (7–12) | 10 (7–14) | **0.005** | 90 (9.5) |
| Alanine aminotransferase (ALT) (U/l) | 32 (22–49) | 30 (21–47) | 34 (23–51) | **0.003** | 46 (4.8) |
| Aspartate aminotransferase (AST) (U/l) | 44 (31–66) | 38 (28–56) | 50 (35–74) | <**0.001** | 173 (18.2) |
| Alkaline phosphatase (U/l) | 70 (56–87) | 73 (58–87) | 69 (55–88) | 0.754 | 111 (11.7) |
| Gamma glutamyl transpeptidase (gGT) (U/l) | 50 (30–88) | 46 (29–82) | 53 (30–95) | **0.024** | 49 (5.1) |
| Lactate dehydrogenase (LDH) | 357 (278–466) | 315 (250–402) | 408 (322–527) | <**0.001** | 91 (9.6) |
| Procalcitonin (μgram/l) | 0.16 (0.08–0.35) | 0.11 (0.06–0.20) | 0.23 (0.11–0.48) | <**0.001** | 393 (41.2) |
| Serum creatinine (μmol/l) | 84 (67–107) | 83 (65–104) | 85 (69–110) | 0.093 | 26 (2.7) |
| **Radiology on admission**–*no. (%)* | | | | | |
| Chest X-ray | 570 (59.9) | 280 (58.8) | 290 (61.1) | - | 169 (17.8)[b] |
| Chest CT scan | 489 (51.4) | 259 (54.4) | 218 (48.5) | - | 169 (17.8)[b] |
| Median CT severity score (IQR) | 12 (8–15) | 10 (7–13) | 13 (10–17) | <**0.001** | 614 (64.5) |
| CORADS score no./total no. of patients (%) | | | | 0.259 | 610 (64.1) |
| CORADS 1 | 13/342 (3.8) | 6/181 (3.3) | 7/161 (4.3) | - | - |
| CORADS 2 | 8/342 (2.3) | 5/181 (2.8) | 3/161 (1.9) | - | - |
| CORADS 3 | 23/342 (6.7) | 14/181 (7.7) | 9/161 (5.6) | - | - |
| CORADS 4 | 38/342 (11.1) | 27/181 (15.0) | 11/161 (6.8) | - | - |
| CORADS 5 | 260/342 (76.0) | 129/181 (71.3) | 131/161 (81.3) | - | - |
| **Treatment during hospital stay**–*no. (%)* | | | | | |
| Chloroquine | 664 (69.7) | 285 (59.9) | 379 (79.6) | <**0.001** | 0 (0) |
| Hydroxychloroquine | 42 (4.4) | 18 (3.8) | 24 (5.0) | 0.344 | 0 (0) |
| Remdesivir | 10 (1.1) | 0 (0) | 10 (2.1) | **0.001** | 0 (0) |
| Lopinavir/Ritonavir | 18 (1.9) | 6 (1.3) | 12 (2.5) | 0.153 | 0 (0) |
| Anakinra | 17 (1.8) | 0 (0) | 17 (3.6) | <**0.001** | 0 (0) |
| None | 245 (25.7) | 171 (35.9) | 74 (15.5) | <**0.001** | 0 (0) |

Abbreviations: COVID-19, coronavirus disease 2019; IQR, interquartile range; CT, computed tomography; CORADS, COVID-19 Reporting and Data System.

[a] P values indicate differences between moderately and severely ill patients. P < 0.05 was considered statistically significant.

[b] Radiology on admission has not been done (both chest X-ray and chest CT-scan).

## Complications

In severely ill patients, the three most common complications that occurred more frequently than in moderately ill patients were: acute kidney injury (AKI; 23.7% versus 12.4%), delirium (17.4% versus 8.0%) and atrial fibrillation (13.9% versus 4.1%), respectively (Table 3). In

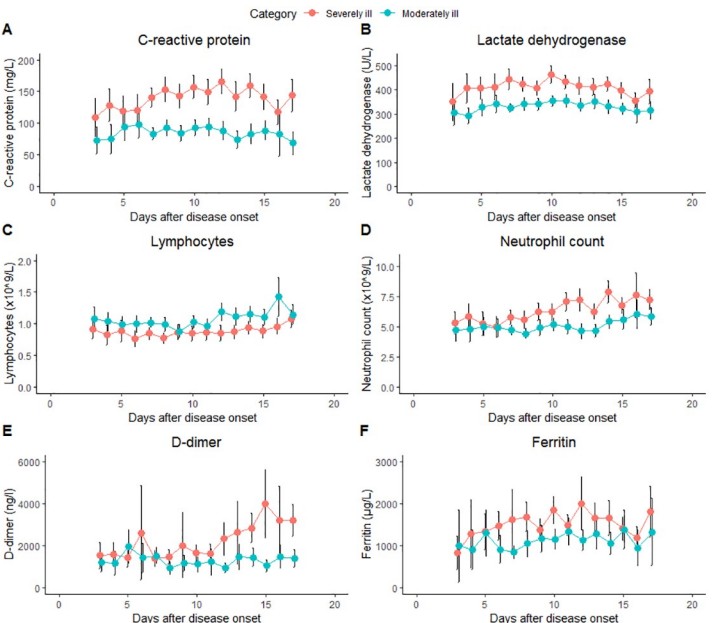

**Fig 1. Repeated measurements of six different laboratory parameter from onset of symptoms until discharge between moderately ill and severely ill COVID-19 patients.** Repeated measurements of six different laboratory parameters illustrate differences between moderately ill patients (blue line) and severely ill patients (red line). All the observations are shown from day 3 after disease onset until day 17. Data are shown as means with 95% confidence intervals. Lymphocyte counts are excluded for 3 patients, because they had chronic lymphatic leukemia.

severely ill patients admitted to ICU, AKI was present in 38.3%, a pulmonary embolism in 23.5% and septic shock in 5.5%. However, in severely ill patients admitted to the general ward, only 10 (3.4%) patients had a pulmonary embolism.

### Clinical outcomes

Table 3 shows the clinical outcomes in the total population: clinical improvement (708 [74.4%]), death (239 [25.1%]) or discharged with palliative care (5 [0.5%]). Of the 952 patients, 183 (19.2%) patients were admitted to ICU after a median of 1 day (IQR 0–4) after admission. Fifty-six (30.6%) of these patients died. The median length of hospital stay was 6 days [IQR 3–9] for severely ill patients admitted to the general wards compared to 26 days [IQR 17–41] for patients admitted to the ICU. Of the 293 severely ill patients admitted to general wards, 139 (47.4%) died during hospitalization. Among the 239 patients that deceased in the entire cohort, 153 (64.0%) had a non ICU admission policy. At the ICU, most patients received invasive ventilation 129 (70.5%) and otherwise non-invasive ventilation 57 (31.1%).

In different age groups (<60 years, 61–69 years and >70 years), in-hospital mortality was 5.8%, 14.1% and 41.0%, respectively. Survival in the different age groups is depicted (S2 Fig in S1 Data).

Among patients admitted to ICU aged >70 years, 50.2% deceased at the ward with a non ICU admission policy compared to 48.4% in patients admitted to ICU (Table 4). The median age of those patients was 80 years (IQR 76–84) at the general ward compared to 74 years (IQR 73–78) at ICU. Among patients aged >70 years that survived, the median age was 80 years (IQR 75–83) at the ward and 72 years (IQR 71–75) at ICU. The median length of ICU stay was 22 days (IQR 9–34) in patients aged >70 years and 21 days (IQR 13–32) in patients aged

**Table 3. Clinical outcomes and in-hospital complications.**

| | Total patients | Moderately ill patients | Severely ill patients | | | P Value[a] |
|---|---|---|---|---|---|---|
| | Total Patients (n = 952) | Total moderately ill patients (n = 476) | Total severely ill patients (n = 476) | Admissions to ward (n = 293) | Admissions to ICU (n = 183) | |
| **Clinical outcome**–*no. (%) or no./total no. of patients (%)* | | | | | | |
| Gender, male | 605 (63.6) | 280 (58.8) | 325 (68.3) | 191 (65.2) | 134 (73.2) | **0.002** |
| Median age:–*(IQR)* | 69 (58–77) | 69 (56–78) | 70 (61–77) | 74 (65–80) | 67 (58–72) | 0.113 |
| *<60 years* | 274 (28.8) | 157 (33.0) | 117 (24.6) | 57 (19.5) | 60 (32.8) | **0.004** |
| *61–69 years* | 205 (21.5) | 92 (19.3) | 113 (23.7) | 54 (18.4) | 59 (32.3) | 0.098 |
| *>70 years* | 473 (49.7) | 227 (47.7) | 246 (51.7) | 182 (62.1) | 64 (35.0) | 0.218 |
| Admitted to ICU | 183 (19.2) | 0 (0) | 183 (38.4) | 0 (0) | 183 (100) | - |
| Non ICU admission policy: | 309 (32.5) | 147 (30.9) | 165 (34.7) | 165 (56.3) | 0 (0) | - |
| *<60 years* | 11/274 (4.0) | 6/157 (3.8) | 5/117 (4.3) | 5/117 (4.3) | 0 (0) | - |
| *61–69 years* | 29/205 (14.1) | 10/92 (10.9) | 19/113 (16.8) | 19/113 (16.8) | 0 (0) | - |
| *>70 years* | 269/473 (56.9) | 131/227 (57.7) | 138/246 (56.1) | 138/246 (56.1) | 0 (0) | - |
| Median no. of days hospitalized before ICU admission (IQR) | 1 (0–4) | 0 (0) | 1 (0–4) | 0 (0) | 1 (0–4) | - |
| Median no. of days at ICU (IQR) | 16 (9–26) | 0 (0) | 16 (9–26) | 0 (0) | 16 (9–26) | - |
| Median length of hospital stay[b] (IQR) | 6 (3–12) | 5 (3–7) | 9 (4–23) | 6 (3–9) | 26 (17–41) | **<0.001** |
| Discharge reason from hospital: | | | | | | |
| *Clinical improvement* | 708 (74.4) | 429 (90.1) | 279 (58.6) | 150 (51.2) | 127 (69.4) | - |
| *Deceased* | 239/952 (25.1) | 44/476 (9.2) | 195/476 (41.0) | 139/293 (47.4) | 56/183 (30.6) | - |
| *<60 years* | 16/239 (6.7) | 3/44 (6.8) | 13/195 (6.7) | 3/139 (2.2) | 10/56 (17.9) | - |
| *61–69 years* | 29/239 (12.1) | 1/44 (2.3) | 28/195 (14.4) | 13/139 (9.4) | 15/56 (26.8) | - |
| *>70 years* | 194/239 (81.2) | 40/44 (90.9) | 154/195 (79.0) | 123/139 (88.5) | 31/56 (55.4) | - |
| *Palliative care* | 5 (0.5) | 3 (0.6) | 2 (0.4) | 5 (1.7) | 0 (0) | - |
| Non ICU admission policy/deceased (%) | 153/239 (64.0) | 32/44 (72.7) | 121/195 (62.1) | 121/139 (87.1) | 0 (0) | - |
| **Ventilation**–*no. (%)* | | | | | | |
| Non-invasive | 97 (10.2) | 11 (2.3) | 86 (18.1) | 29 (9.9) | 57 (31.1) | - |
| Invasive | 129 (13.6) | 0 (0) | 129 (27.1) | 0 (0) | 129 (70.5) | - |
| ECMO | 1 (0.1) | 0 (0) | 1 (0.2) | 0 (0) | 1 (0.5) | - |
| **Complications**–*no. (%)* | | | | | | |
| COPD or asthma exacerbation | 14 (1.5) | 4 (0.8) | 10 (2.1) | 7 (2.4) | 3 (1.6) | 0.106 |
| Pulmonary embolism | 61 (6.4) | 8 (1.7) | 53 (11.1) | 10 (3.4) | 43 (23.5) | **<0.001** |
| Acute kidney injury | 172 (18.1) | 59 (12.4) | 113 (23.7) | 43 (14.7) | 70 (38.3) | **<0.001** |
| Liver function abnormalities | 92 (9.7) | 27 (5.7) | 65 (13.7) | 21 (7,2) | 44 (24.0) | **<0.001** |
| Atrial fibrillation | 87 (9.1) | 21 (4.4) | 66 (13.9) | 27 (9.2) | 39 (21.3) | **<0.001** |
| Ventricular arrythmia | 8 (0.8) | 2 (0.4) | 6 (1.3) | 1 (0.3) | 5 (2.7) | 0.156 |
| Cardiogenic shock | 2 (0.2) | 0 (0) | 2 (0.4) | 0 (0) | 2 (1.1) | 0.157 |
| Myocarditis | 7 (0.7) | 0 (0) | 7 (1.5) | 4 (1.4) | 3 (1.6) | **0.008** |
| Heart failure | 23 (2.4) | 6 (1.3) | 17 (3.6) | 12 (4.1) | 5 (2.7) | **0.020** |
| Hyperglycemia | 50 (5.3) | 16 (3.4) | 34 (7.1) | 12 (4.1) | 22 (12.0) | **0.009** |
| Hypoglycemia | 13 (1.4) | 9 (1.9) | 4 (0.8) | 4 (1.4) | 0 (0) | 0.163 |
| Cerebrovascular accident | 17 (1.8) | 4 (0.8) | 13 (2.7) | 4 (1.4) | 9 (4.9) | **0.028** |
| Septic shock | 11 (1.2) | 0 (0) | 11 (2.3) | 1 (0.3) | 10 (5.5) | **0.001** |

*(Continued)*

**Table 3.** (Continued)

| | Total patients | Moderately ill patients | Severely ill patients | | | P Value[a] |
|---|---|---|---|---|---|---|
| | Total Patients (n = 952) | Total moderately ill patients (n = 476) | Total severely ill patients (n = 476) | Admissions to ward (n = 293) | Admissions to ICU (n = 183) | |
| Delirium | 121 (12.7) | 38 (8.0) | 83 (17.4) | 25 (8.5) | 58 (31.7) | <**0.001** |

Abbreviations: COVID-19, coronavirus disease 2019; ICU, intensive care unit; IQR, interquartile range; ECMO, extracorporeal membrane oxygenation; COPD, chronic obstructive pulmonary disease.

[a] P values indicate differences between moderately and severely ill patients. P < 0.05 was considered statistically significant.

[b] Length of hospital stay starts from the first day of admission until discharge or death.

**Table 4. Characteristics and outcomes of patients stratified by age (<60, 61–69 and >70), split in subgroups by ICU admission policies or ICU admission and mortality.**

| | Patients aged 70 years or older (n = 473) | | | | | |
|---|---|---|---|---|---|---|
| | Admitted to ward with no ICU restrictions (n = 140) | | Admitted to ward with non ICU admission policy (n = 269) | | Admitted to ICU (n = 64) | |
| **Characteristic—n (%)** | Survived 112 (80.0%) | Deceased 28 (20.0%) | Survived 134 (49.8%) | Deceased 135 (50.2%) | Survived 33 (51.6%) | Deceased 31 (48.4%) |
| Median age—(IQR) | 74 (72–78) | 78 (75–81) | 80 (75–83) | 80 (76–84) | 72 (71–75) | 74 (73–78) |
| Gender, male | 77 (68.8) | 21 (75.0) | 82 (61.2) | 86 (63.7) | 22 (66.7) | 27 (87.1) |
| Severely ill | 27 (24.1) | 17 (60.7) | 32 (23.9) | 106 (78.5) | 33 (100) | 31 (100) |
| Median length of hospital stay (IQR) | 5 (3–9) | 5 (4–7) | 7 (4–10) | 5 (3–7) | 31 (26–49) | 17 (8–24) |
| Median length of ICU stay (IQR) | 0 (0) | 0 (0) | 0 (0) | 0 (0) | 22 (9–34) | 12 (6–22) |
| | Patients aged 61 to 69 years or older (n = 205) | | | | | |
| | Admitted to ward with no ICU restrictions (n = 117) | | Admitted to ward with non ICU admission policy (n = 29) | | Admitted to ICU (n = 59) | |
| **Characteristic—n (%)** | Survived 116 (99.1%) | Deceased 1 (0.9%) | Survived 16 (55.2%) | Deceased 13 (44.8%) | Survived 44 (74.6%) | Deceased 15 (25.4%) |
| Median age—(IQR) | 66 (63–68) | 65 (65–65) | 67 (64–68) | 67 (65–69) | 67 (64–68) | 67 (65–67) |
| Gender, male | 64 (55.2) | 1 (100) | 9 (56.3) | 8 (61.5) | 32 (72.7) | 10 (66.7) |
| Severely ill | 34 (29.3) | 1 (100) | 7 (43.8) | 12 (92.3) | 44 (100) | 15 (100) |
| Median length of hospital stay (IQR) | 5 (3–9) | 8 (8–8) | 9 (4–14) | 5 (3–7) | 23 (35–52) | 15 (4–28) |
| Median length of ICU stay (IQR) | 0 (0) | 0 (0) | 0 (0) | 0 (0) | 21 (13–32) | 13 (4–26) |
| | Patients aged 60 years or under (n = 274) | | | | | |
| | Admitted to ward with no ICU restrictions (n = 203) | | Admitted to ward with non ICU admission policy (n = 11) | | Admitted to ICU (n = 60) | |
| **Characteristic—n (%)** | Survived 202 (99.5%) | Deceased 1 (0.5%) | Survived 6 (54.5%) | Deceased 5 (45.6%) | Survived 50 (83.3%) | Deceased 10 (16.7%) |
| Median age—(IQR) | 52 (48–56) | 49 (49–49) | 56 (49–58) | 57 (55–59) | 53 (47–58) | 56 (48–59) |
| Gender, male | 115 (56.9) | 0 (0) | 4 (66.7) | 4 (80.0) | 35 (70.0) | 8 (80.0) |
| Severely ill | 52 (25.7) | 0 (0) | 2 (33.3) | 3 (60.0) | 50 (100) | 10 (100) |
| Median length of hospital stay (IQR) | 4 (3–7) | 3 (3–3) | 6 (5–10) | 9 (2–12) | 25 (17–39) | 25 (17–30) |
| Median length of ICU stay (IQR) | 0 (0) | 0 (0) | 0 (0) | 0 (0) | 14 (9–21) | 23 (12–26) |

Abbreviations: ICU, intensive care unit; IQR, interquartile range.

between 61 and 69 years. In patients aged <60 years the median length of ICU stay was 14 days (IQR 9–21), In general, ICU admitted patients seemed to have less comorbidities in all age categories compared to patients admitted to the ward with a non ICU admission policy (S1 Table in S1 Data). Furthermore, we visualized the survival of patients aged 70 years and over based on ICU admission policy (S3 Fig in S1 Data). Patients aged 70 years and over with a non ICU admission policy show similar survival after 30 days compared to patients of this age group admitted to the ICU. Moreover, patients above 70 years deceased early during the course of hospital admission as compared to patients in this age group who were admitted to the ICU."

## Discussion

This multicenter cohort study, including 952 hospitalized patients with a PCR-confirmed SARS-CoV-2 infection in the Netherlands, is one of the largest multicenter cohort studies of COVID-19 in Northern Europe. COVID-19 in hospitalized patients in The Netherlands proved to be a disease of older age and the prevalence of comorbidities were high in this population, congruent with the findings of a large observational study in Europe [21]. The median age in this cohort was 69 years. Similarly, male patients and patients with cardiovascular disease were overrepresented [4, 10]. Sex affected disease severity on presentation whereas other patient characteristics including age, obesity and immunocompromised state, did not. However, striking differences in inflammatory parameters and D-dimer were observed upon admission and during hospitalization between the moderately and severely ill patients.

Previous research shows that D-dimer may have prognostic associations [22]. Our results show indeed that D-dimer is fairly more increased at admission in the severely ill patients. Moreover, our results show that D-dimer is rising during hospital stay in severely ill patients, where there seems to be no increase of D-dimer in moderately ill patients. This has been previously reported for survivors versus non-survivors and can be seen in light of increased inflammatory and coagulation status during prolonged illness [23]. This also might be related to a higher prevalence of pulmonary embolism especially in ICU patients [24]. Lymphopenia and a higher neutrophil-to-lymphocyte ratio on admission were seen in severely ill patients, which is consistent with literature reporting that these measurements are considered to be an independent risk factor for mortality [25].

Complications were mostly observed in patients admitted to ICU. Previous research showed a cumulative incidence of thrombotic complications of 31% of ICU patients, consisting of 81% who had a pulmonary embolism [24]. In our study, 23.5% of patients admitted to ICU developed pulmonary embolism. Of note, much less pulmonary embolisms were found in the severely ill patients admitted to general wards. We hypothesize that this is due to less frequent diagnostic work-up for pulmonary embolism in this subset of patients. AKI was observed mostly in patients admitted to ICU (38.3%), corresponding with previous studies [26].

In our study, the in-hospital mortality was 25.1%. Among patients aged 70 years and over, 41% died. Reported in-hospital mortality varies in literature between 4.3% and 28% [4, 9, 23]. It is known that increased age is associated with mortality in COVID-19 [23] and also observed in large databases [27]. Reported mortality during ICU admission is also highly variable (between 26% and 78%) in countries across the world [4, 7, 28]. This observed variety in ICU mortality rates of COVID-19 patients could potentially be due to heterogeneity in populations characteristics and institutional policies [29], but possibly also differences concerning end-of-life issues. In the PRoVENT-COVID study, also conducted in The Netherlands and with an ICU population with matching patient characteristics (age and gender distribution), ICU mortality rates were roughly equal [30].

This cohort is especially interesting compared to other cohorts described so far since many patients aged >70 years, had a non ICU admission policy. This signifies that ICU admission was considered too much burden for the patient given factors such as fragility, comorbidities or patients' own disposition. Weighing the potential benefits and risk in ICU triage for patients being considered for ICU admission is important [31], but it is difficult to determine which COVID-19 patients will thrive with or without ICU care. In addition, little evidence is available about which patients in general will benefit most from ICU admission [29].

In this manuscript, we demonstrate similar survival rates between patients above 70 years admitted to the ICU compared to patients in the same age group with a non ICU admission policy. In the latter group many patients deceased early during the course of their hospital admission, thereby possibly preventing patient burden that would have occurred if they were admitted to an ICU. Knowledge of patient characteristics, clinical outcomes, prognostic factors and course of disease are required to more distinctively identify factors that could help to differentiate which COVID-19 patients benefit from ICU admission.

Our lower ICU mortality may be explained by the fact that almost a third of all patients had a non-ICU admission policy, resulting in a relative more vital population admitted to the ICU. Although patients >70 at the ICU were relatively younger compared to patients >70 admitted to the ward with a non ICU admission policy, mortality rates were equally high (around 50%). Also, ICU survival improved when patients were aged <60 years. Furthermore, the median length of ICU stay was 8 days longer in patients aged >70 years, possibly leading to more complications. Previous research showed that poor mental health and functional disability were common and persistent in patients recovering from ICU [32], as well as physical morbidity in patients admitted to ICU with acute lung injury [33]. Moreover, a large amount of COVID-19 patients report at least 1 ongoing symptom after hospital discharge [34]. Finally, an increasing length of ICU stay is associated with long-term mortality rates in elderly hospital survivors [35].

Taken into account abovementioned factors, the patient burden of an ICU admission, possible long term complications after discharge, longer length of ICU stay and the high mortality in patients aged >70 years should encourage us to further investigate the meaningfulness of ICU admissions in elderly and fragile COVID-19 patients. When ICU capacity is scarce caused by an international health crisis such as COVID-19, this will be even more difficult, but inevitable to provide the best healthcare possible for (fragile) elderly COVID-19 patients.

Strengths of this study include its multicenter design and the large number of included patients. It is one of the largest cohort studies to report on outcomes of COVID-19 without patients still being hospitalized. However, this study also has some limitations. First, this was an observational, retrospective study where data were retrieved from electronical medical records. Second, there was missing data for some patients, especially some laboratory values. Third, follow up of our included patients stopped after discharge or death, resulting in small assumptions in our survival curve. Finally, because of the multicenter approach, differences in manner of reporting and clinical management could have occurred, but also increased the generalizability of findings in this study.

## Conclusion

To the best of our knowledge, this is one of the largest cohort studies in Europe among COVID-19 patients, in which definite clinical outcomes can be described. A large percentage of elderly patients in this cohort had a non ICU admission policy. Hospitalized COVID-19 patients aged >70 years had high mortality and a large group of patients aged >70 years were admitted to the ward because of a non ICU admission policy. This group had similar mortality

rates compared to patients aged >70 years who were admitted to the ICU. Mortality occurred earlier in the course of hospital admission in the first group, reflecting the possible benefit of being spared a high burden of disease which comes with an ICU admission. The length of ICU stay was longer in patients aged >70 years who survived ICU admission. These findings in combination with the patient burden of an ICU admission and possible long term complications after discharge should encourage us to further investigate the benefit of ICU admission in elderly and fragile COVID-19-patients.

## Supporting information

**S1 Data.**
(DOCX)

## Acknowledgments

We thank all students and nurses involved with extracting data from electronic records. Furthermore, we thank Mr Ruben Bosch for his assistance with the statistics used in this report.

## Author Contributions

**Conceptualization:** Josephine van de Maat, Karin Veerman, Jaap ten Oever, Anton Dofferhoff, Jacobien Hoogerwerf.

**Data curation:** Josephine van de Maat, Nico Janssen, Evertine Abbink.

**Formal analysis:** Niels Pouw.

**Investigation:** Niels Pouw, Josephine van de Maat, Karin Veerman, Jaap ten Oever, Nico Janssen, Evertine Abbink, Monique Reijers, Quirijn de Mast, Wouter Hoefsloot, Reinout van Crevel, Kitty Slieker, Marjan van Apeldoorn, Marc Blaauw, Anton Dofferhoff, Jacobien Hoogerwerf.

**Methodology:** Niels Pouw, Josephine van de Maat, Karin Veerman, Jaap ten Oever, Anton Dofferhoff, Jacobien Hoogerwerf.

**Project administration:** Niels Pouw, Josephine van de Maat, Nico Janssen, Jacobien Hoogerwerf.

**Software:** Niels Pouw.

**Supervision:** Josephine van de Maat, Anton Dofferhoff, Jacobien Hoogerwerf.

**Visualization:** Niels Pouw.

**Writing – original draft:** Niels Pouw, Karin Veerman, Jacobien Hoogerwerf.

**Writing – review & editing:** Niels Pouw, Josephine van de Maat, Karin Veerman, Jaap ten Oever, Nico Janssen, Evertine Abbink, Monique Reijers, Quirijn de Mast, Wouter Hoefsloot, Reinout van Crevel, Kitty Slieker, Marjan van Apeldoorn, Marc Blaauw, Anton Dofferhoff, Jacobien Hoogerwerf.

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
