## [Decision Letter · Decision Letter 0]

14 Jan 2021

PONE-D-20-40220

Clinical Characteristics and outcomes of 952 hospitalized COVID-19 patients in The Netherlands: a retrospective cohort study

PLOS ONE

Dear Dr. Hoogerwerf,

Thank you for submitting your manuscript to PLOS ONE. After careful consideration, we feel that it has merit but does not fully meet PLOS ONE’s publication criteria as it currently stands. Therefore, we invite you to submit a revised version of the manuscript that addresses the points raised during the review process.

We look forward to receiving your revised manuscript.

Kind regards,

Francesco Di Gennaro

Academic Editor

PLOS ONE

Journal Requirements:

2. In the Methods, please clarify that participants provided oral consent. Please also state in the Methods:

- Why written consent could not be obtained

- Whether the Institutional Review Board (IRB) approved use of oral consent

- How oral consent was documented

For more information, please see our guidelines for human subjects research: https://journals.plos.org/plosone/s/submission-guidelines#loc-human-subjects-research

3. In the ethics statement in the manuscript and in the online submission form, please provide additional information about the patient records/samples used in your retrospective study, including:

a) whether all data were fully anonymized before you accessed them;

b) the date range (month and year) during which patients' medical records/samples were accessed.

6. Please include captions for your Supporting Information files at the end of your manuscript, and update any in-text citations to match accordingly. Please see our Supporting Information guidelines for more information: http://journals.plos.org/plosone/s/supporting-information

Additional Editor Comments:

dear Authors follow reviewer suggestion to improve your paper

Reviewers' comments:

Reviewer's Responses to Questions

**Comments to the Author**

1. Is the manuscript technically sound, and do the data support the conclusions?

Reviewer #1: Yes

Reviewer #2: Yes

2. Has the statistical analysis been performed appropriately and rigorously? 

Reviewer #1: Yes

Reviewer #2: I Don't Know

3. Have the authors made all data underlying the findings in their manuscript fully available?

Reviewer #1: Yes

Reviewer #2: Yes

4. Is the manuscript presented in an intelligible fashion and written in standard English?

Reviewer #1: Yes

Reviewer #2: Yes

5. Review Comments to the Author

Reviewer #1: I read with great interest the manuscript. Authors wrote a very good article.

Only some minor suggestion

1.Introduction: update data on global burden of SARS CoV2 infection

2. Methods and result are clear. Well tables and statistical analysis

3. DIscussusion: discuss better the pharmacological approach also in other European countries ( see and citeUse of hydroxychloroquine in hospitalised COVID-19 patients is associated with reduced mortality: Findings from the observational multicentre Italian CORIST study. Eur J Intern Med. 2020 Dec;82:38-47. doi: 10.1016/j.ejim.2020.08.019. Epub 2020 Aug 25. PMID: 32859477; PMCID: PMC7446618 and RAAS inhibitors are not associated with mortality in COVID-19 patients: Findings from an observational multicenter study in Italy and a meta-analysis of 19 studies. Vascul Pharmacol. 2020 Dec;135:106805. doi: 10.1016/j.vph.2020.106805. Epub 2020 Sep 28. PMID: 32992048; PMCID: PMC7521934.. Compare your data with comorbidites in other big data article Common cardiovascular risk factors and in-hospital mortality in 3,894 patients with COVID-19: survival analysis and machine learning-based findings from the multicentre Italian CORIST Study. Nutr Metab Cardiovasc Dis. 2020 Oct 30;30(11):1899-1913. doi: 10.1016/j.numecd.2020.07.031. Epub 2020 Jul 31. PMID: 32912793 )

Furthermore add data on policy and lockdown during pandemic

Reviewer #2: Pouw and colleagues are to be commended for their accurate ISARIC-compliant descriptive work on 952 patients with coronavirus 2 disease. Although the findings are clearly reported and an interesting longitudinal follow up is performed on several variables, here are some few points that need to be addressed:

Major

1. It is not clear if the study was registered

2. Two groups are presented in the statistical anlysis (severe vs. moderate), but the reason for analysing based on these two groups should be stated earlier, and also moderate disease should be defined.

3. Of the patients admitted to ICU it is not clear how many received invasive or non invasive ventilation

4. The one third of patients with no-ICU admission policy is presented as a key finding, and it is actually the finding that makes this paper more interesting compared to other retrospective cohorts. Althouth TAble 4 and Supp. table 1 try to cover this I believe the characteristics of the patients that chose to go for a non-icu admission could be better exposed. E.g. did they present with different vital signs? The interesting finding that above 70yrs mortality is the same is also important but seems undersnowed now. Also, no mention to the non-ICU admission policy is made in the paper conclusions. E.g. A figure like S2 but considering groups based on ICU admission policy would be interesting. I am not suggesting to conduct data-driven reporting, just that a salient finding compared to many others cohort studies be better explored and conveyed.

5. Two groups are shown in figure 1, but it is not clear if a longitudinal comparison wants to be tested (then should be added to methods). e.g. is CRP consistently different among the groups or not? when saying that Ddimer changes after 15 days, no hypothesis was tested there, should we only eyeball?

6. Although for some reason received treatment is not an endpoint of the study, it would help to know major things like use of oxygen and NIV in the two groups.

7. In discussion the findings related to ICU admissions could be compared to the findings of the PRoVENT-COVID study performed in the same country, to see if the findings from the hospitals in the two regions participating in the study are comparable to the wider country.

8. Very few study limitations are presented.

9. For longitudinal data very little info is provided on how it was collected ("every 48-72h" is rather imprecise and hampers replicability). And for vital signs collected more than once a day, were the worse used, or the best, etc.? Please add some details for clarity. Also, vital signs collected are not reported anywere it seems.

Minor

line 182: please reword (repetitive use of "most common", "more frequently")

6. PLOS authors have the option to publish the peer review history of their article (what does this mean?). If published, this will include your full peer review and any attached files.

Reviewer #1: No

Reviewer #2: No

---

## [Author Response · Author response to Decision Letter 0]

6 Feb 2021

In the attached file named Response to Reviewers the specific reviewer and editor comments are discussed.

---

## [Decision Letter · Decision Letter 1]

4 Mar 2021

Clinical Characteristics and outcomes of 952 hospitalized COVID-19 patients in The Netherlands: a retrospective cohort study

PONE-D-20-40220R1

Dear Dr. Jacobien Hoogerwerf,

We’re pleased to inform you that your manuscript has been judged scientifically suitable for publication and will be formally accepted for publication once it meets all outstanding technical requirements.

Kind regards,

Francesco Di Gennaro

Academic Editor

PLOS ONE

Additional Editor Comments (optional):

dear authors congratulations

Reviewers' comments:

Reviewer's Responses to Questions

**Comments to the Author**

1. If the authors have adequately addressed your comments raised in a previous round of review and you feel that this manuscript is now acceptable for publication, you may indicate that here to bypass the “Comments to the Author” section, enter your conflict of interest statement in the “Confidential to Editor” section, and submit your "Accept" recommendation.

Reviewer #1: All comments have been addressed

Reviewer #2: All comments have been addressed

2. Is the manuscript technically sound, and do the data support the conclusions?

Reviewer #1: Yes

Reviewer #2: Yes

3. Has the statistical analysis been performed appropriately and rigorously? 

Reviewer #1: Yes

Reviewer #2: Yes

4. Have the authors made all data underlying the findings in their manuscript fully available?

Reviewer #1: Yes

Reviewer #2: Yes

5. Is the manuscript presented in an intelligible fashion and written in standard English?

Reviewer #1: Yes

Reviewer #2: Yes

6. Review Comments to the Author

Reviewer #1: Authors wrote an very good article. They improved their already good version and now can be accept.

Reviewer #2: Thank you for addressing all comments in a detailed and ordered way. I think the manuscript improved in the process and the message is better conveyed.

7. PLOS authors have the option to publish the peer review history of their article (what does this mean?). If published, this will include your full peer review and any attached files.

Reviewer #1: No

Reviewer #2: **Yes: **Luigi Pisani

---

## [Editor Report · Acceptance letter]

9 Mar 2021

PONE-D-20-40220R1 

Clinical characteristics and outcomes of 952 hospitalized COVID-19 patients in The Netherlands: a retrospective cohort study 

Dear Dr. Hoogerwerf:

I'm pleased to inform you that your manuscript has been deemed suitable for publication in PLOS ONE. Congratulations! Your manuscript is now with our production department. 

Kind regards, 

on behalf of

Dr. Francesco Di Gennaro 

Academic Editor

PLOS ONE